# Stability of Graphene/Nafion Composite in PEM FC Electrodes

**DOI:** 10.3390/nano14110922

**Published:** 2024-05-24

**Authors:** Anna O. Krasnova, Nadezhda V. Glebova, Angelina G. Kastsova, Anna O. Pelageikina, Alexey V. Redkov, Maria V. Tomkovich, Andrey A. Nechitailov

**Affiliations:** 1Department of Solid State Electronics, Ioffe Institute, 194021 St. Petersburg, Russia; glebova@mail.ioffe.ru (N.V.G.); akastsova@mail.ioffe.ru (A.G.K.); a.o.pelagejkina@mail.ioffe.ru (A.O.P.);; 2Institute for Problems in Mechanical Engineering, Russian Academy of Sciences, 199178 St. Petersburg, Russia; rav@ipme.ru

**Keywords:** PEM FC electrodes, stability, nafion, graphene, ORR

## Abstract

Ensuring the stable operation of proton exchange membrane fuel cells is conducive to their real-world application. A promising direction for stabilizing electrodes is the stabilization of the ionomer via the formation of surface compounds with graphene. A comprehensive study of the (electrochemical, chemical, and thermal) stability of composites for fuel cell electrodes containing a modifying additive of few-layer graphene was carried out. Electrochemical stability was studied by cycling the potential on a disk electrode for 5000 cycles. Chemical stability was assessed via the resistance of the composites to H_2_O_2_ treatment using ion-selective potentiometry. Thermal stability was studied using differential thermal analysis. Composites were characterized by UV-Vis spectroscopy, Raman spectroscopy, EDX, and SEM. It was shown that graphene inhibits Nafion degradation when exposed to heat. Contrariwise, Nafion is corrosive to graphene. During electrochemical and chemical exposure, the determining change for carbon-rich composites is the carbon loss (oxidation) of the carbon material. In the case of carbon-poor composites, the removal of fluorine and sulfur from the Nafion polymer with their partial replacement by oxygen prevails. In all cases, the F/S ratio is stable. The dispersity of Nafion in a sample affects its chemical stability more than the G/Nafion ratio does.

## 1. Introduction

Fuel cells [1] and water electrolyzers [2,3] with a proton exchange membrane (PEM) are gaining popularity for a number of reasons: the absence of significant carbon emissions into the environment, high-energy conversion efficiency, autonomy, low noise, and control flexibility. These devices are based on the principle of converting the energy of chemical bonds into electrical energy and vice versa (Figure 1). One of the key fundamental properties of the electrochemical electrodes of these devices is mixed conductivity. Electronic conduction is responsible for the transfer of electrons from the anode through an external electrical circuit to the cathode. Proton conduction is responsible for the transfer of hydrated protons within the membrane electrode assembly from the anode to the cathode through the PEM. Mixed conductivity is ensured by the presence of two types of materials: electron-conducting (carbon materials, metals) and proton-conducting (Nafion) materials. The problem of increasing the service life of these devices largely determines their cost, commercial attractiveness, and competitiveness [4,5]. The durability problem consists of many components and remains relevant to this day. Thus, during the operation of these devices, their components are strongly affected due to local overheating [6,7], chemical and electrochemical corrosion [8,9], and mechanical stresses [10]. Degradation affects not only the main components of the membrane electrode assembly (MEA), such as the metal catalyst, carrier, membrane, and ionomer in the layer, but also structural components: current plates, seals, and so forth. For example, Zhao et al. [11] reviewed the degradation of sealing materials; Leng et al. [12] studied the corrosion of bipolar stainless steel plates.

In relation to ionomer (in the form of a membrane and as an ionomer distributed in the electrodes) degradation, works aimed at stabilizing the membranes are presented. Thus, two main directions can be distinguished: reinforcement and filling. The creation of various coatings in the form of polymer networks increases the service life of Nafion due to various stabilization mechanisms [13,14], and that of coatings in the form of matrices [15] provides a longer service life and greater mechanical strength for Nafion-based membranes. Jao et al. [16] noted that the advantages of a polytetrafluoroethylene (PTFE)/Nafion composite membrane are its low cost, high mechanical strength, and relatively low degree of swelling. This study focuses on the properties of (PTFE)/Nafion membranes and (PTFE)/Nafion MEAs by comparing the durability and performance characteristics of (PTFE)/Nafion MEAs with those of commercial Nafion 211 MEAs. Shi et al. [13] highlighted that durability still poses a technical barrier to the commercialization of perfluorosulfonic acid membranes (most commonly Nafion), which are currently used in PEM fuel cells.

The use of fillers to create composite (filled) membranes has proven to be a way to increase resistance to water loss and thermal stability [17,18,19,20,21,22,23]. Teixeira et al. [24] proposed new modified Nafion membranes doped with bisphosphonic acids with increased resistance to chemical degradation by H_2_O_2_/Fe^2+^ (Fenton’s reagent), simulating ex situ radical attack on the membrane structure. The new membranes showed excellent chemical stability after oxidative degradation under Fenton test conditions at 80 °C and durability compared with that of commercial Nafion 115 membranes. Agarwal et al. [25] noted that the introduction of radical scavenger additives, such as cerium, is a promising solution to the problem of membrane destruction. One of the directions for membrane stabilization is the creation of membranes based on more stable ionomers [26,27,28]. Kononova et al. [27] showed the prospects of polyimidesulfonic acid BDSA-SPI-4 (H) as a proton-conducting membrane polymer for electrochemical devices with direct energy conversion; a maximum electrical power of ~170 mW/cm^2^ was reported in fuel cell mode in an oxygen/hydrogen system at room temperature and atmospheric pressure.

A relatively new direction is the use of carbon materials in Nafion proton-conducting membranes to improve their characteristics, such as moisture content at elevated temperatures, ionic resistance (conductivity), and thermal stability [18,19,20]. For this purpose, additives such as carbon nanotubes [20,21] and graphene-like materials [20,21,22] are used.

The thermal stability of PEM fuel cells and their long-term operation are the subjects of numerous studies, for instance [29,30,31,32,33,34,35,36,37,38]. One of the promising approaches to increasing the stability of the ionomer is the use and strengthening of the interface interaction of components, the creation of adsorption and chemical surface compounds to improve corrosion resistance and structural stability, and the expansion of the operating temperature range [33,34,35,36,37,38]. In these works, the interaction of the proton exchange polymer Nafion with the surface of various carbon materials, carbon black, multi-walled carbon nanotubes, graphene, and platinum, was studied. The strongly expressed stabilizing effect of carbon materials with a large surface area (graphene) on the thermal stability of the polymer was demonstrated. Nafion–carbon interfaces were studied using nuclear magnetic resonance and X-ray photoelectron spectroscopy methods, showing strong interaction at the level of formation of surface chemical compounds.

Thus, the literature indicates that the degradation of the ionomer in the form of a membrane or the ionomer distributed in the electrode occurs in a number of directions, often interconnected. The most significant are the following: mechanical destruction, chemical destruction as a result of a radical attack, thermal destruction associated with dehydration and structural destruction, colloidization, and migration of the ionomer in an electric field. It should be noted that the ionomer in the electrode is less susceptible to mechanical destruction compared with the membrane ionomer.

The described approaches for stabilizing the ionomer in the MEA can be grouped as follows: (1) mechanical stabilization by reinforcement with related fluorinated polymers; (2) thermodynamic stabilization based on the creation of stable composites and surface compounds; (3) kinetic stabilization using free radical scavengers that reduce their concentration; (4) structural stabilization using composites based on various nanostructured materials, including carbon ones, to increase moisture independence and thermal stability, and control the morphology and porosity of the electrode to suppress electrophoretic migration.

A promising direction for the stabilization of the ionomer in the electrode is the formation of surface compounds with graphene. Yet, the resistance of the graphene/Nafion (G/Nafion) composite system to various types of exposure has not been sufficiently studied.

The study and use of G in various fields of technology has recently become increasingly popular due to its unique properties [39,40,41]. Zeng et al. [39] described principles for controlling electromagnetic waves using a G surface. Jiang et al. [40] outlined the use of G in photoelectronics: in a photovoltaic device for photoelectric and electrically induced thermo-optical conversions. Han et al. [41] described a thermoelectric system based on G, and studied the thermoelectric properties of armchair-type G nanoribbons formed as a result of partial adsorption of hydrogen and fluorine atoms.

The purpose of this work is to comprehensively study the resistance of the G/Nafion composite to various types of influence: electrochemical, chemical, and thermal.

## 2. Materials and Methods

### 2.1. Composites and MEA Preparation

Thermally expanded graphite (TEG) was used for G fabrication [42] (NPO “Graphene Materials”, St. Petersburg, LLC, Russia). The technology of TEG synthesis is described in [43]. E-TEK platinized carbon black (40% Pt) [44], Nafion solution DE1021 (DuPont™, Wilmington, DE, USA), isopropanol (99.80%, ECOS-1 JSC), and deionized water with a resistivity at room temperature of ρ ≥ 18 MOhm × cm were used.

Two types of composites were prepared: Nafion-TEG without Pt with various ratios of components: 1:4, 1:1, 4:1; Nafion-TEG-Pt/C with various ratios of Nafion-TEG: 1:4, 1:2, 1:1. The mass fraction of Pt/C was 30% for the composites containing Pt (Table 1). Composites were fabricated via mechanical mixing followed by ultrasonic homogenization of the initial components. For this purpose, precisely measured amounts of materials were mixed in a polyethylene vessel, and a mixture of isopropanol and water in a volume ratio of 1:1 was added. The solid-to-liquid ratio in the dispersion was maintained in the range of 1:1040–1:4000 for composites without Pt and 1:410–1:440 for composites with Pt, depending on the content of TEG. The higher the TEG content, the more of the liquid phase was added to ensure complete wetting of the solid components. The vessel was placed in a Branson 3510 ultrasonic bath (Branson Ultrasonics Corporation, Danbury, CT, USA). Ultrasonic treatment was carried out at an operating frequency of *f* = 40 kHz and a power of 130 W for ~40–100 h in order to obtain a homogeneous dispersion that remained stable without separation for at least 1 min.

The composites and pristine Nafion ionomer were deposited on the glossy side of silicon substrates measuring 22 × 14 mm^2^ at a temperature of *T* = 70–80 °C using a dispenser. The substrates were pre-treated in isopropanol for 15 min and washed with deionized water. The amount of deposited material (monitored gravimetrically) was in the range of 1.75 ± 0.08 mg. The samples were used for UV-Vis spectroscopy, chemical stability studies, energy-dispersive X-ray spectroscopy (EDX), and Raman spectroscopy.

MEAs were fabricated by depositing a homogeneous dispersion of components directly on a PEM through a stainless-steel mask. Prior to the deposition of the electrode material, the membrane was treated in 3% H_2_O_2_ solution for 10 min at a temperature of 70–80 °C, then was kept in 0.5 M H_2_SO_4_ for 15 min at 70–80 °C and then was washed five times with deionized water. The electrodes were prepared by depositing prepared composite dispersions containing Pt on a Nafion-type PEM, Nafion N-212, (DuPont, Wilmington, DE, USA) with a thickness of 50 μm. The membrane was thermostated at 85 °C on an Ika C-MAG HP 7 wafer (IKA-Werke GmbH & Co. KG, Staufen, Germany) with a heat controller. The amount of the deposited material was monitored gravimetrically.

### 2.2. UV-Vis Spectroscopy

UV-Vis spectra were recorded using a Specord 210 (Analytik Jena, Jena, Germany) spectrophotometer with a rate of 5 nm/s, a resolution of 0.1 nm, and a split of 1 nm. The spectra were measured in a quartz cell with an absorbing layer measuring 1 cm long; as the reference solution, the isopropanol + water (1:1) mixture was used.

Dispersions for the study were prepared by diluting the initial dispersions as follows: 2 drops (0.08 mL) of prepared dispersions were diluted with 4 mL of the isopropanol + water (1:1) mixture for dispersions without Pt and with 13 mL of the isopropanol + water (1:1) mixture for dispersions with Pt.

### 2.3. Differential Thermal Studies

Differential thermal analysis was carried out using the Mettler Toledo TGA/DSC 1 derivatograph with the STARe System software V16.40 (Mettler-Toledo LLC, Columbus, OH, USA), blowing air through the derivatograph chamber at a rate of 30 cm^3^/min in the mode of uniform temperature rise at a rate of 10 K/min from 35 to 1000 °C. A few milligrams of the material was placed in an alund crucible, and mass (thermogravimetric (TG)) and thermal (differential thermal (DT)) curves were recorded during heating.

Dispersions for the study were evaporated, poured onto a glass substrate, and dried at a.c. to an air-dried state (~40–50% relative humidity), and then the dry residue was removed with a metal (stainless-steel) spatula. Composites of the compositions given in Table 1 were subjected to this study.

### 2.4. Study of the Resistance of Composites to Radicals

Composites on silicon substrates and the pristine Nafion ionomer were exposed to hydrogen peroxide, followed by measuring the amount of fluoride ions released during the chemical destruction of Nafion via direct potentiometry using a fluoride-selective electrode. For this purpose, substrates with applied layers were kept in a 10% H_2_O_2_ solution (50 mL) at a temperature of 80 °C for 30 min. The sample was then removed, and platinized platinum (Pt/Pt) was placed in the remaining solution to decompose the remaining hydrogen peroxide. Pt/Pt was kept in solution until the release of oxygen bubbles stopped. The remaining solution was brought to 50 mL with deionized water, and 10 mL of a buffer of constant ionic strength was added. The potential was then measured using a fluoride-ion selective electrode. To carry this out, a fluoride-selective electrode of the ELIS-131F brand (LLC “Izmiritelnaya Tekhnika”, Moscow, Russia) and a silver chloride electrode of the Sr-10101 brand (LLC “Izmiritelnaya Tekhnika”, Moscow, Russia) were placed in the resulting solution. The measurement was carried out using a P-40X potentiostat (Elins LLC, Chernogolovka, Russia). The concentration of fluoride ions was determined from a calibration curve.

To prepare 1 dm^3^ of the buffer of constant ionic strength, precisely weighed portions of the following substances (of at least analytical grade) were taken: NaCl (58.5 g), CH_3_COOH (15.0 g), CH_3_COONa·3H_2_O (102.0 g), and sodium citrate (0.26 g).

Calibration solutions were prepared using sodium fluoride, NaF, and using the serial dilution method to plot the calibration curve. Potentials were measured alternately from minimum to maximum concentrations of calibration solutions using a P-40X potentiostat, the fluoride-selective electrode of the ELIS-131F brand, and the silver chloride electrode of the Sr-10101 brand.

Composites of the compositions given in Table 1 and pristine Nafion were subjected to this study.

### 2.5. RDE Studies

Four types of objects were studied: Nafion-TEG composites with various ratios of components: 1:4, 1:1, and 4:1; pristine Nafion polymer. The above-mentioned sample dispersions were applied to the glassy carbon surface (0.07 cm^2^) of the disk electrode and air-dried for approximately 1 h.

A three-electrode cell with a silver chloride reference electrode and a graphite counter electrode in 0.5 M sulfuric acid in equilibrium with air at 25 °C, IPC-Pro MF potentiostat (LLC “NTF “Volta”, St. Petersburg, Russia), and VED-06 setup (LLC “NTF “Volta” St. Petersburg, Russia) were used for electrochemical studies. The electrochemical impedance method was used to record the ohmic resistance of the solution. The resistance measured according to the high-frequency cutoff of the hodograph along the axis of real values was in the range of 7–8 Ω.

Potentiometry, cyclic voltammetric analysis, and potentiostatic amperometry methods were used to examine the samples on the disk electrode. The samples were exposed to potential cycling at a rate of 50 mV/s between 0.05 and 1 V vs. RHE for 5000 cycles. The equilibrium electrode potential, double layer charging current (*I*_EDL_), and oxygen reduction reaction (ORR) current (*I*_ORR_) were measured between cycles. The ORR current was measured at a fixed potential of 0.05 V vs. RHE and a rotational speed of RDE of ω = 2000 rpm.

### 2.6. Raman Spectroscopy

Nafion-TEG composites were studied using a confocal Raman microscope, Witec Alpha 300R (Oxford Instruments, Ulm, Germany), equipped with a 532 nm excitation laser (output power of ~30 mW). The spectra were collected from the pristine Nafion-TEG composites placed on the silicon substrate and from the Nafion-TEG composites after H_2_O_2_ treatment. Each spectrum was accumulated 3 times for 15 s and then averaged. For peak detection, the spectra were subjected to Savitzky–Golay smoothing. The baseline was calculated using an OriginLab^®^ implementation of asymmetric least squares smoothing. The peak positions were calculated using an OriginLab^®^ implementation.

### 2.7. Scanning Electron Microscopy

A target-oriented approach was used to optimize analytic measurements [45]. Before measurements, the samples were mounted on a 3 mm copper grid and fixed in a grid holder. The samples’ morphology was studied using a Hitachi SU8000 (Hitachi High-Tech Corp., Tokyo, Japan) field-emission scanning electron microscope (FE-SEM). Images were acquired in bright-field STEM mode at a 30 kV accelerating voltage.

### 2.8. Energy-Dispersive X-ray Spectroscopy

Samples both before and after chemical and electrochemical exposure were studied. EDX measurements were carried out using the FEI Quanta 200 scanning electron microscope (FEI Company, Hillsboro, OR, USA).

### 2.9. MEA Testing

For testing the composites as a part of MEA electrodes, the MEA was placed in a standard two-electrode measuring cell (FC-05-02, ElectroChem Inc., Woburn, MA, USA) with graphite current collectors [46]. Toray 090 standard carbon paper was used as the gas diffusion layer.

Before performing electrochemical measurements, MEAs were kept in 0.5 M H_2_SO_4_ for 1 h at *T* = 70–80 °C, washed five times with deionized water, placed in a testing cell, and then activated as described in [47]. The current–voltage characteristic (CVC), H_2_ crossover current density, and electrochemically active surface area (ESA) of Pt were measured in accordance with the DOE test protocol for cell performance [48] using P-40X potentiostat (Elins LLC, Chernogolovka, Russia). The CVCs were registered at a sweep rate of *v* = 10 mV/s under O_2_/H_2_ flow, at *T* = 25 °C and atm. pressure. The H_2_ crossover through the membrane was assessed via the reduction current of hydrogen passing through the membrane in an N_2_/H_2_ environment. The ESA of Pt was measured via cyclic voltammetry under N_2_/H_2_ flow through hydrogen desorption in accordance with a well-known method [1,49] based on the measurement of the charge passed for hydrogen desorption from the platinum surface.

### 2.10. Calculation and Processing of Results

The Pt ESA was calculated using the following ratio:*S*_Pt_ = *Q*_des_/210(1)
where *S*_Pt_—Pt ESA, cm^2^; *Q*_des_—charge spent on hydrogen desorption, µC; 210—specific charge spent on hydrogen desorption from the surface of Pt, µC/cm^2^.

Pt loading was gravimetrically calculated using the following ratio:*G*_Pt_ = *M*_CL_ × *N* × 0.4(2)
where *G*_Pt_—Pt loading in the electrode, mg/cm^2^; *M*_CL_—catalytic layer weight, mg; *N*—proportion of Pt/C in the catalytic layer; 0.4—proportion of Pt in E-TEK.

The Pt specific activity (SA) in the ORR was calculated using the following ratio:SA *=* (*J*@*E* × *S*_CL_)/*S*_Pt_(3)
where SA—Pt SA at a potential of *E* = 0.9 V vs. RHE, mA/cm^2^ (ESA Pt); *J*@*E*—current density of the ORR at a potential of *E* = 0.9 V vs. RHE, mA/cm^2^; *S*_CL_—visible electrode surface area, *S*_CL_ = 1 cm^2^; *S*_Pt_—Pt ESA, cm^2^.

The polarization capacitance to build the CVC was calculated using the following ratio:*C*_EDL_ = *I*/*v*(4)
where *C*_EDL_—polarization capacitance of the electrical double layer (EDL), F; *I*—current, mA; *v*—potential sweep rate, mV/s.

The fraction of fluorine that went into the solution in the form of F^−^ when exposed to hydrogen peroxide was calculated using the following ratio:*m*_F_/*m*_Nafion_ = (*C*_F_ × 19 × *V*_sample_)/(*m* × ω_Nafion_)(5)
where *C*_F_—measured concentration of F^−^ in the solution, gram-ion/dm^3^; *V*_sample_—sample volume, (0.05 dm^3^); *m*—sample mass, g; ω_Nafion_—fraction of Nafion in the sample.

## 3. Results

Figure 2 shows microphotographs of the Nafion-TEG(1:4) composite. As can be seen from the figure, the material consists of particles with a characteristic size of approximately 50 μm (Figure 2a). At higher magnification (Figure 2b,c), the fine structure of the material’s particles, consisting of stacks and sheets of G, becomes visible.

Figure 3 illustrates the absorption spectra of the dispersion of Nafion-TEG various compositions. The spectrum exhibits two maximums: 230 nm, which corresponds to the π−π* transition of *sp*^2^ C=C bonds, and ∼265 nm, which corresponds to graphene oxide (GO) reduced to G [50]. Lai et al. [51] demonstrated that dispersions containing lowlayer (1−3 layers) GO could be distinguished from dispersions containing multilayer (4−10 layers) GO and GO with a higher number of layers (>10) by the intense 230 nm peak in their UV-Vis spectra. When the number of layers increases (>10), the intensity of the 230 nm peak tends to drop off.

### 3.1. Differential Thermal Analysis

Figure 4 shows derivatograms of the studied materials: Nafion-TEG without Pt and Nafion-TEG-Pt/C. The derivatograms of all studied samples contain three characteristic temperature intervals, which are responsible for the occurrence of various physical and chemical processes of material destruction [34,52,53,54]. The lowest-temperature region is responsible for the evaporation of water; in the medium-temperature region, the destruction of Nafion occurs; and in the high-temperature region, combustion of carbon material transpires.

In the case of a two-component composition (Figure 4a), the following temperature ranges are observed: 1. an area of evaporation of water (35–110 °C); 2. a region of oxidative destruction of Nafion (300–514 °C); 3. a TEG oxidation region (514–720 °C).

The characteristic temperature ranges of the destruction of three-component composites (Figure 4b) are shifted towards low temperatures, which is associated with the presence of platinum [34].

In the case of three-component composites, the following temperature ranges are observed: 1. a water evaporation region (35–110 °C); 2. a region of oxidative destruction of Nafion (280–423 °C); 3. a TEG oxidation region (423–705 °C).

The thermal stability of the composites was assessed using the shift in the peaks of Nafion destruction and TEG destruction to the region of lower or higher temperatures. It can be seen from the derivatograms of two-component and three-component composites that as the proportion of G increases, the temperature of destruction of Nafion also increases. This results from the more complete distribution of Nafion on the surface of G sheets [37]. The presence of platinum does not affect this trend; it only contributes to a shift in the characteristic peaks of differential thermogravimetric (DTG) curves to the low-temperature region. Simultaneously, the temperature of the peak on the DTG curve representing the maximum oxidation rate of carbon material varies with changes in the G/Nafion ratio. The smaller the G fraction, the lower the peak temperature. This can be explained by the fact that G is more in contact with Nafion when the latter is in excess. After the destruction of the polymer adsorbed on the surface of G, the carbon material acquires an additional number of structural defects, which reduces its thermal stability.

To prove the validity of this assumption, Raman spectra of the initial Nafion-TEG(1:1)-Pt/C composite and the material remaining after heating that composite on a derivatograph to a temperature of *T* = 424 °C followed by rapid cooling were obtained. Figure 5 illustrates the Raman spectra of these samples.

The figure shows that both spectra have the same qualitative characteristics. The studied carbon flakes are samples of few-layer G, the Raman spectra of which contain three characteristic bands: *D*, *G*, and 2*D*. The *D*′ band contributes to the slight asymmetry of the *G* band. The presence of line *D* and line *D*′ in the spectra indicates the presence of defects in the structure of the material. The 2*D* line is well approximated by a single Lorentz function (see insets), which is typical for graphene [55]. At the same time, due to the presence of defects in its crystal lattice, the 2*D* line has a rather large width (FWHM ~83–85 cm^−1^), which is characteristic of few-layer G containing 2–3 monolayers. Analysis of the spectra shows that both before and after heat treatment of the composite, it contains G; however, a more detailed analysis of the quantitative characteristics of the spectra (Table 2) suggests that the defectiveness of G after heat treatment increases. The increase in the defectiveness of G is indicated by a significant boost in the ratios of intensities and areas under the peaks of the *D* and *G* lines, some broadening of the *G* line (FWHM(*G*) increased from 30 to 36 cm^−1^), and a more pronounced *D*′ line. Meanwhile, the shape of the 2*D* line remained virtually unchanged. The Raman spectra of the pristine Nafion-TEG(1:1)-Pt/C material, along with those obtained following thermal treatment, are summarized in Table 2.

### 3.2. Electrochemical Exposure on RDE

Figure 6 demonstrates the dynamics of cyclic voltammograms of the samples studied during the process of repeated potential application.

The figures demonstrate that the CVCs of all samples are slightly inclined relative to the potential axis. As is known, this is caused by the presence of atmospheric oxygen in the electrolyte solution; its reduction occurs in the cathode region. The CVCs of all samples are in similar current ranges: ~0.02–0.05 mA. In the case of TEG-containing samples, the first CVCs show peaks in the cathodic region (485–531 mV) and response peaks in the anodic (425–445 mV) region at the beginning of electrochemical action. Upon the subsequent application of a cyclic potential, these peaks disappear. These peaks are likely caused by the presence of surface compounds on the G sheets. Also, all composites containing TEG have peaks at a potential of approximately 300 mV, associated with the quinone–hydroquinone equilibrium on the G surface [56].

Figure 7 illustrates the dependences of the electrochemical characteristics of the materials studied during potential cycling.

The change in equilibrium potential (Figure 7a) during electrochemical action for samples not rich in carbon material occurs to an insignificant extent. For a sample with a high TEG content (Nafion-TEG(1:4)), a rather substantial decrease in potential is observed at the beginning, followed by a plateau. Upon comparing the values (relative positions of the curves) of the equilibrium potential (Figure 7a) and the capacitance of the EDL (Figure 7c) for samples of various compositions, it is clear that the order of the *C*_EDL_(*N*) curves is the reverse of the order of the *E*(*N*) curves.

The nature of the dependence of the stationary current of the ORR on the number of potential cycles (Figure 7b) is determined by the composition of the material. In the case of the composite with the largest proportion of TEG (Nafion-TEG(1:4)), there is a dependence with the maximum and highest current values. For other compositions, the dependences are approximated by linear equations, while the magnitude of the ORR current changes slightly during the electrochemical action. The absolute values of the ORR current are also determined by the TEG content. In the case of pristine Nafion, the ORR currents are the smallest (approximately 1 μA), and they are apparently associated with the influence of the glassy carbon surface of the electrode on which the Nafion film is deposited. The ORR currents are highest (4–8 μA) in the case of the TEG-rich sample (Nafion-TEG(1:4)). The magnitude of the ORR current correlates with the TEG fraction in the composite, with the general rule being that the larger the TEG fraction, the greater the current. The maximum at ~2000 cycles in the dependence of the current on the number of cycles (Figure 7b—red curve) could be associated with the processes of formation and destruction of oxygen-containing groups of atoms on the surface of the carbon material and with an increase in the area of the electrochemically active surface of the carbon material (Figure 7c). However, this phenomenon requires a separate study. It should be noted that the absolute values of the ORR current are significantly (more than an order of magnitude) lower thanthe value of the limiting diffusion current at ω = 2000 rpm for both four-electron and two-electron mechanism (0.102 mA for an electrode with *S* = 0.07 cm^2^ and for *n* = 4 ē) [57]; therefore, the ORR is in the area of reaction control.

It can be concluded that the electrochemical characteristics of the studied composites, such as the equilibrium potential, the ORR current, and the capacitance of the EDL of composites with a relatively small TEG content change slightly during electrochemical exposure. A more notable alteration in characteristics is noted for a composite with high TEG content. The proportion of TEG in the composite determines both the level (value) of its characteristics and the law of its change under electrochemical action.

### 3.3. Chemical Stability (Resistance to Hydrogen Peroxide)

Table 3 shows data on the amount of fluoride ions that went into solution during chemical treatment with hydrogen peroxide. Data are presented for samples of various compositions.

The table reveals that as the G/Nafion ratio decreases, the fraction of fluorine that passes into the solution also decreases. The observed trend is noted for both series with and without Pt. The smallest fraction of fluorine passes into the solution during the treatment of pristine Nafion (0.12). It can be concluded that the fraction of fluorine transferred into the solution is similar when samples with similar compositions with and without Pt (Nafion-TEG(1:1)-Pt/C) and Nafion-TEG(1:1)) are compared. As the proportion of Nafion in the sample decreases, the proportion of fluorine that passes into the solution increases (Figure 8). This could be associated with the influence of dispersity on the reaction kinetics. In the case of a low content of ionomer distributed in the material, its surface area in contact with reagents (hydrogen peroxide and its decomposition products, radicals) is significantly greater than that of a bulk ionomer. The dependence of the fraction of fluorine transferred into the solution on the Nafion content in the sample in the studied range is approximated by a second-degree polynomial.

Based on the data collected, it can be concluded that the dispersity of Nafion in the sample affects its chemical stability more than the G/Nafion ratio does.

Table 4 and Figure 9 present the results of a study on changes in the elemental composition of samples during treatment with hydrogen peroxide and electrochemical exposure.

The table demonstrates that all samples have some variation in composition upon exposure. However, the variations in the case of pristine Nafion and in the case of composites occur differently.

Thus, after electrochemical exposure to Nafion, the most significant changes are an increase in the proportion of carbon, a decrease in the proportion of fluorine, and an increase in the proportion of oxygen. The ratio of the F/S mass percent ratio changed slightly after electrochemical action. At the same time, the C/F mass percent ratio increased from 0.6 to 0.8. The change in the ratio indicates the preferential removal of fluorine and sulfur from the Nafion polymer as a result of the degradation of the composition with their partial replacement by oxygen. The oxygen content, as shown in the table, increased slightly from 7.07 to 9.1%, and the F/O ratio decreased.

For composite materials, different trends are observed. As with Nafion, there is relative stability in the F/S ratio. The C/F ratio for compositions rich in carbon materials (G and carbon black), both containing and not containing platinum (Nafion-TEG(1:4)-Pt/C and Nafion-TEG(1:4), respectively), drops significantly after any kind of exposure. For a carbon-poor composition (Nafion-TEG(4:1)), the C/F ratio remains constant. The dominant process that determines the change in composition for composites rich in carbon materials is the process of loss (oxidation) of carbon. In the case of composites poor in carbon materials, the determining process is the destruction of Nafion.

Figure 10 illustrates the Raman spectra of composites of the two-component G/Nafion system with various ratios of components; Table 5 summarizes the result of mathematical processing of the Raman spectra of composites of the system Nafion-TEG without Pt of various compositions before and after exposure to hydrogen peroxide.

The results in Table 5 show that all samples exhibit a rise in the defectiveness of few-layer G after H_2_O_2_ treatment. Along with this, the degree of disordering of the G structure does not discernibly depend on the composition of the composite. The full width at half maximum of the 2*D* band (FWHM(2*D*), cm^−1^) is in the range of 80–87 cm^−1^ and does not change noticeably after exposure to hydrogen peroxide.

The Raman spectra of the Nafion-TEG(4:1) sample for the region between 640 and 1410 cm^−1^ are depicted in Figure 11. Raman spectra were normalized by the band intensity of the symmetric stretching vibration of CF_2_, *ν*(C–F), at around 735 cm^−1^.

The stretching vibrations of CF_2_, SO, CO, and CS bands of the Nafion backbone and side chains were observed around 1217, 1063, 974, 807, and 673 cm^−1^, respectively. The C–C single bond peaks were observed around 1300 and 1382 cm^−1^. Peaks observed around 974 cm^−1^ overlap with the silicon peak from the silicon substrate. The features of the Raman spectra for Nafion are consistent with those described in previous studies [58,59,60].

The positions, intensities, and shapes of the Raman bands of the Nafion polymer in each sample did not change significantly, with the exception of those for *ν*(C–S). The constancy of the peak intensities corresponding to the *ν*(S–O) bond (wavenumber 1063 cm^−1^) and the area under them indicates that the equivalent mass of Nafion does not significantly change after H_2_O_2_ treatment [61], which is consistent with the EDX data. At the same time, there is an increase in both the intensity and area under the peaks responsible for the *ν*(C–S) bond (673 and 807 cm^−1^) after H_2_O_2_ treatment. According to [61], the increase in these peaks is associated with an increase in the length of side chains in the polymer molecule. Such a change in Raman spectra in our case may indicate the breakdown of the central carbon chain of the molecule into smaller fragments with relatively longer side chains.

### 3.4. Electrochemical Characteristics

Testing of the MEA in a two-electrode cell showed the presence of high diffusion resistance, expressed by a diffusion bend in the CVCs. The high diffusion resistance could be attributed to the low porosity of electrodes composed of the composites. However, the fabricated MEAs demonstrated SA and H_2_ crossover current densities comparable to those of traditional Nafion 211 and Nafion 212 catalysts and membranes (Figure 12) [62,63,64].

The low value of H_2_ crossover current density (0.33 mA/cm^2^—Nafion-TEG(1:1)-Pt/C) may be associated with the formation of a barrier layer on the membrane surface during the fabrication process of the MEA, which prevents the crossover of molecular hydrogen through the membrane. This effect requires further study.

## 4. Discussion

Three types of effects on the composite material of the Nafion-TEG-Pt/C system were studied: thermal, electrochemical, and chemical. The patterns of changes in composition under various types of influence on materials of various compositions were established.

The study of the thermal effect on composites of the Nafion-TEG-Pt/C system of various compositions showed that the stability of the composite increased with the G content. Destructive processes begin to happen at a higher temperature. The presence of platinum does not affect this trend; it contributes to a shift in the characteristic peaks of DTG curves to the low-temperature region. Concurrently, the temperature of the maximum rate of oxidation of carbon material (peak on the DTG curve) changes. As the content of G shrinks, the peak temperature goes down. This could be explained by the enhanced interaction between G and Nafion due the an excess of the latter. After the destruction of the polymer adsorbed on the surface of G, the carbon material acquires an additional number of structural defects, which reduces its thermal stability. Hence, G exhibits an inhibitory effect on the Nafion degradation reaction. In contrast, Nafion is corrosive to G.

The electrochemical properties, including the equilibrium potential, the ORR current, and the EDL capacitance of composites with a relatively small TEG content change slightly during long-term electrochemical exposure to cyclic potential sweeps in the range of 50–1000 mV vs. RHE. The composite with high TEG content shows a more noticeable change in performance. The proportion of TEG in the composite determines both the level (value) of the characteristics and the law of its change under electrochemical action. With all this, the following composition change occurs. After electrochemical exposure to pristine Nafion, the most significant changes in composition are an increase in the proportion of carbon, a decrease in the proportion of fluorine, and an increase in the proportion of oxygen. The ratio of the F/S mass percent slightly changes after electrochemical action. Fluorine and sulfur are replaced by oxygen.

For composite materials, different trends are observed. As with Nafion, there is relative stability in the F/S ratio. The C/F ratio for carbon-rich materials (G and carbon black), both with and without Pt, drops significantly after exposure. For a carbon-poor composition (Nafion-TEG(4:1)), the C/F ratio remains constant.

Thus, the dominant process that determines the change in composition for composites rich in carbon materials is the process of loss (oxidation) of carbon. In the case of composites poor in carbon materials, the determining process is the destruction of Nafion.

The following patterns emerge upon the chemical exposure of the investigated composite materials to hydrogen peroxide. As the G/Nafion ratio decreases, the fraction of fluorine that goes into the solution also decreases. The observed trend is noted for both series of samples that contain platinum and those that do not. The smallest fraction of fluorine passed into the solution during the treatment of pristine Nafion (0.12). The presence of platinum does not affect the decomposition process. As the proportion of Nafion in the sample decreases, the proportion of fluorine that goes into the solution increases. This could be associated with the influence of dispersity on the reaction kinetics. In the case of a low content of ionomer distributed in the material, its surface area in contact with reagents (hydrogen peroxide and its decomposition products, radicals) is significantly greater than that in the case of compact film of the pristine ionomer. Thus, the dispersity of Nafion in a sample affects its chemical stability more than the G/Nafion ratio does. Under chemical exposure, relative stability of the F/S ratio transpires. The C/F ratio for compositions rich in carbon materials (G and carbon black), both with and without Pt, decreases significantly after exposure to hydrogen peroxide, as in the case of electrochemical exposure.

An investigation of the structure of G using Raman spectroscopy showed an increase in the defectiveness of few-layer G after treating the composite with hydrogen peroxide, which is common for all samples. Along with this, the degree of disordering of the G structure does not discernibly depend on the composition of the composite. The full width at half maximum of the 2*D* band (FWHM(2*D*), cm^−1^) is in the range of 80–87 cm^−1^ and does not change noticeably after exposure to hydrogen peroxide.

## Figures and Tables

**Figure 1 nanomaterials-14-00922-f001:**
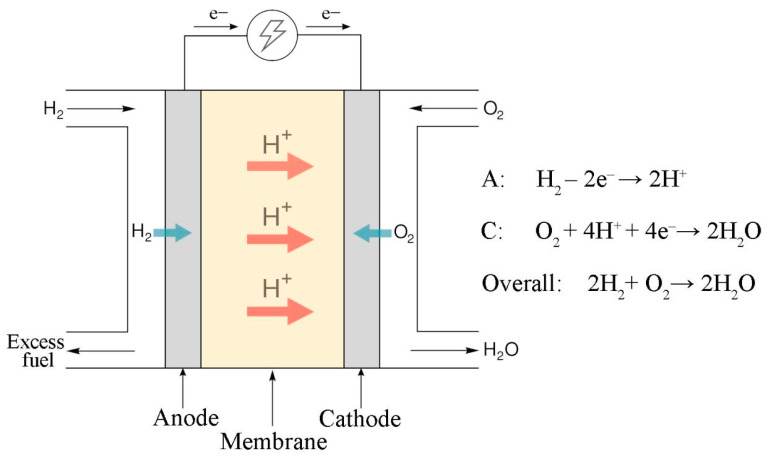
Operating scheme of a PEM fuel cell.

**Figure 2 nanomaterials-14-00922-f002:**
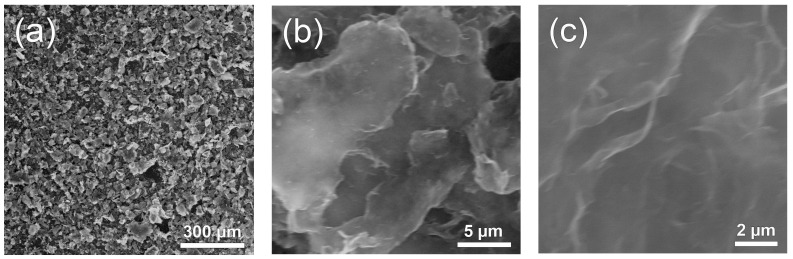
(**a**–**c**) SEM images of initial composite Nafion-TEG (1:4) at various magnifications.

**Figure 3 nanomaterials-14-00922-f003:**
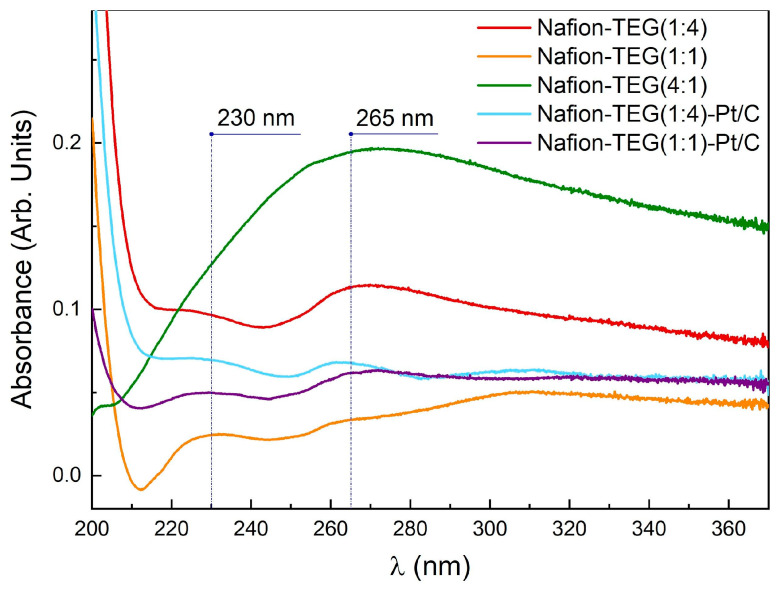
UV-Vis spectra of Nafion-TEG composites.

**Figure 4 nanomaterials-14-00922-f004:**
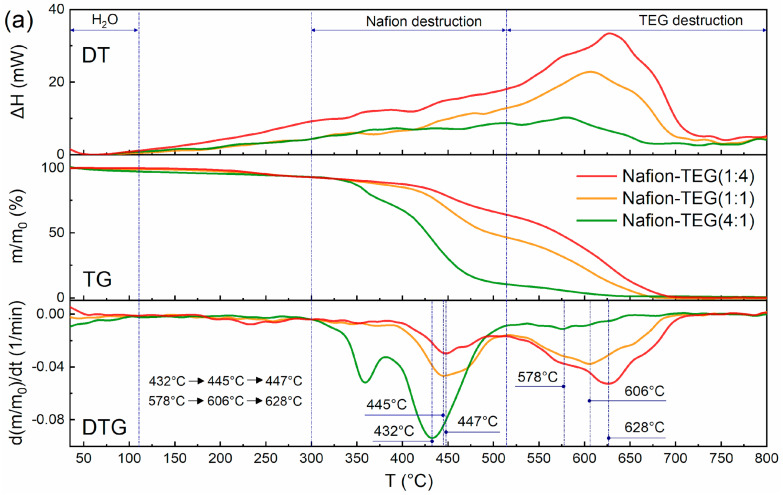
Derivatograms of Nafion-TEG composite materials: (**a**) Nafion-TEG without Pt and (**b**) Nafion-TEG-Pt/C. The heating rate is 10 K/min.

**Figure 5 nanomaterials-14-00922-f005:**
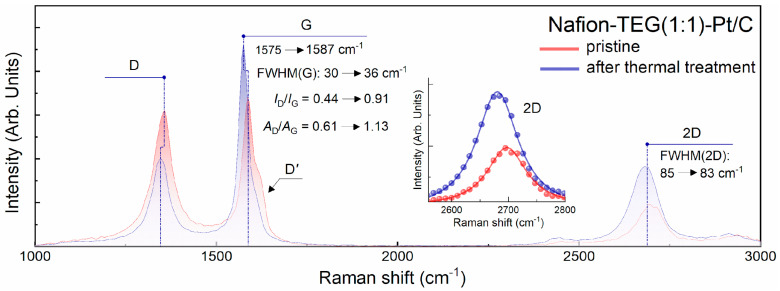
Raman spectra of the Nafion-TEG(1:1)-Pt/C composite: pristine and after thermal treatment under 424 °C.

**Figure 6 nanomaterials-14-00922-f006:**
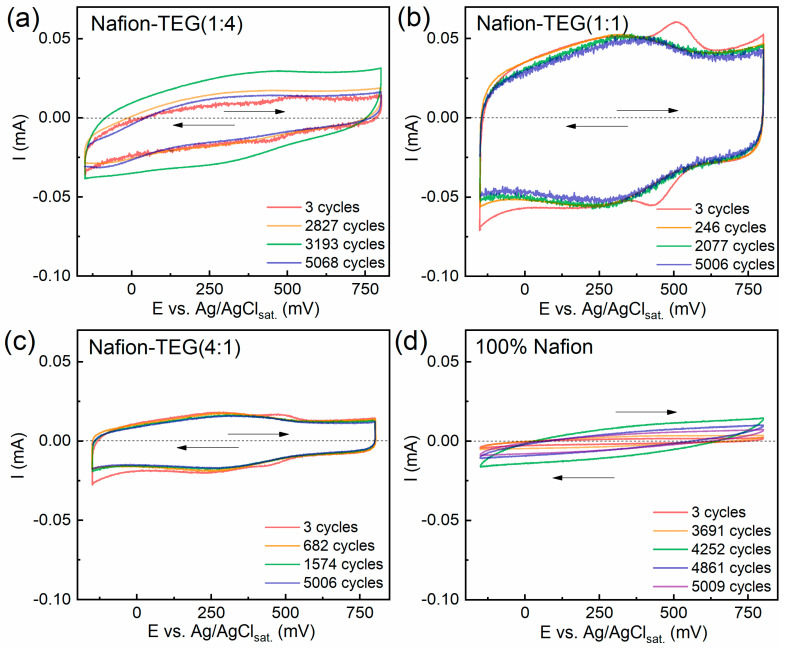
(**a**–**d**) Changes in CVCs of Nafion and Nafion-TEG composites during repeated electrode potential cycling; *T* = 25 °C; polarization rate = 50 mV/s; 0.5 M H_2_SO_4_ in equilibrium with air; arrows indicate polarization directions.

**Figure 7 nanomaterials-14-00922-f007:**
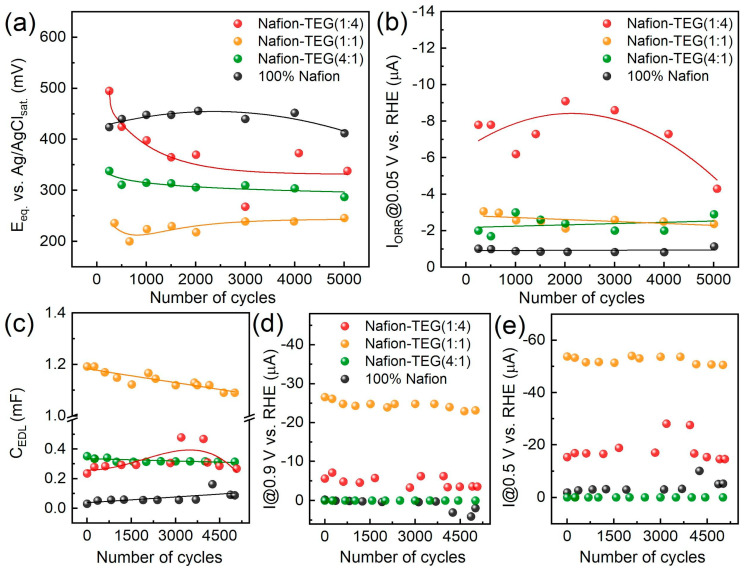
Changes in some properties of Nafion and Nafion-TEG composites during repeated electrode potential cycling; *T* = 25 °C; polarization rate = 50 mV/s: (**a**) change in *E*_eq._; (**b**) change in the ORR current measured at *E* = 0.05 V vs. RHE in potentiostatic mode, with a rotational speed of RDE of ω = 2000 rpm; (**c**) change in *C*_EDL_; (**d**) cathode current measured in potentiodynamic mode at *E* = 0.9 V vs. RHE; (**e**) cathode current measured in potentiodynamic mode at *E* = 0.5 V vs. RHE.

**Figure 8 nanomaterials-14-00922-f008:**
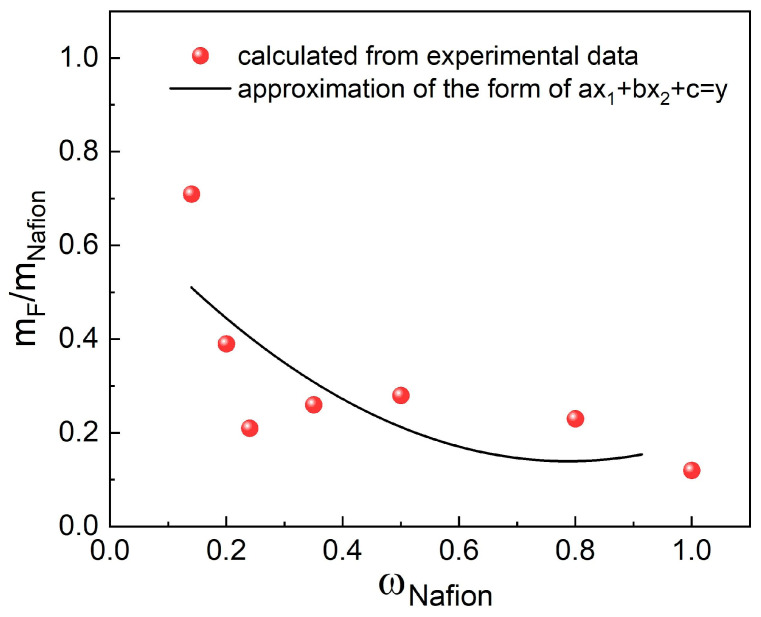
Dependence of the fraction of fluorine that goes into solution after contact with hydrogen peroxide on the Nafion content.

**Figure 9 nanomaterials-14-00922-f009:**
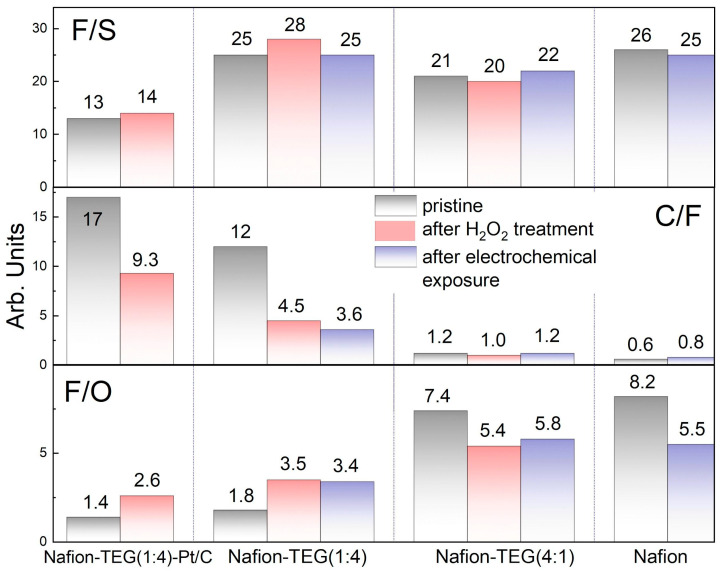
Changes in the ratio of components in the samples after two types of treatment: hydrogen peroxide and electrochemical treatment.

**Figure 10 nanomaterials-14-00922-f010:**
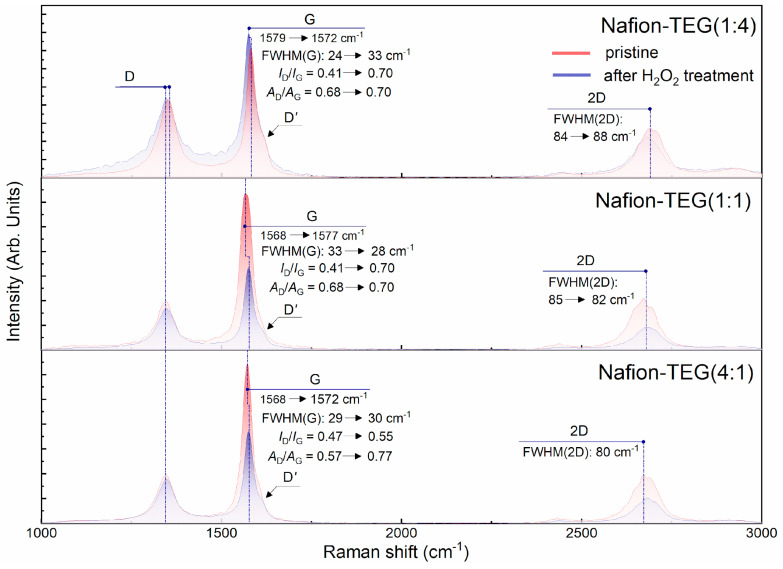
Changes in Raman spectra of Nafion-TEG composites before and after H_2_O_2_ treatment.

**Figure 11 nanomaterials-14-00922-f011:**
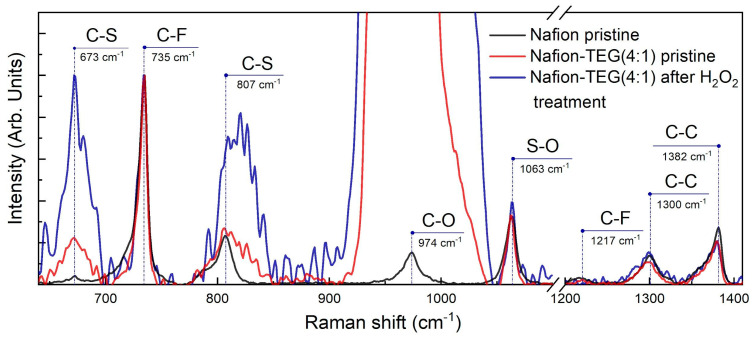
Raman spectra of Nafion-TEG (4:1) before and after H_2_O_2_ treatment and of pristine Nafion.

**Figure 12 nanomaterials-14-00922-f012:**
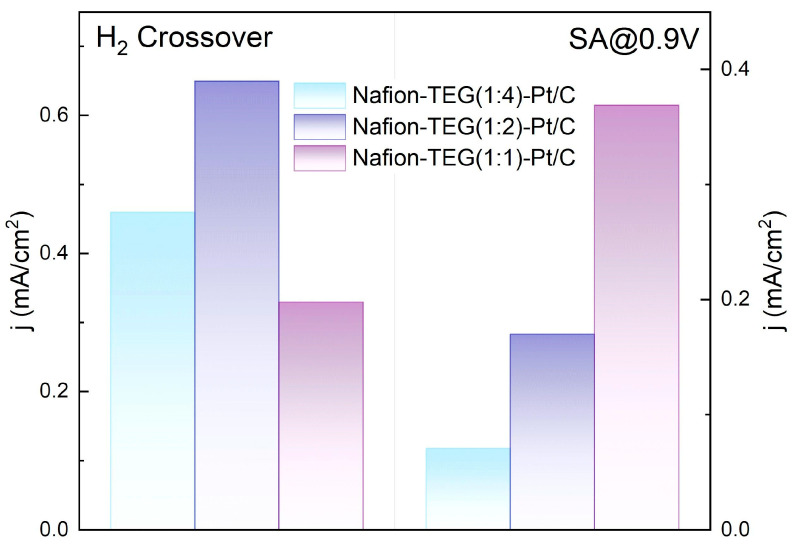
H_2_ crossover current density and SA at 0.9 V vs. RHE of MEAs with Nafion N-212 membrane and electrodes of various compositions.

**Table 1 nanomaterials-14-00922-t001:** Composition of samples used for the study.

Sample	Composition, wt%			MEA Cathode/Anode Pt Loading, mg
	Nafion	TEG	Pt/C (40%Pt)	
Nafion-TEG(1:4)	20	80	0	-
Nafion-TEG(1:1)	50	50	0	-
Nafion-TEG(4:1)	80	20	0	-
Nafion-TEG(1:4)-Pt/C	14	56	30	0.12/0.08
Nafion-TEG(1:2)-Pt/C	24	46	30	0.13/0.09
Nafion-TEG(1:1)-Pt/C	35	35	30	0.09/0.09

**Table 2 nanomaterials-14-00922-t002:** Raman spectra characteristics for the Nafion-TEG(1:1)-Pt/C sample: pristine and after thermal treatment under 424 °C.

Sample	*I* _D_	*I* _G_	*A* _D_	*A* _G_	*I*_D_/*I*_G_	*A*_D_/*A*_G_	FWHM(*G*), cm^−1^	*G*-Band Position, cm^−1^	FWHM(*2D*), cm^−1^
Nafion-TEG(1:1)-Pt/C pristine	41	94	2132	3509	0.44	0.61	30	1575	85
Nafion-TEG(1:1)-Pt/C after thermal treatment	64	70	3501	3086	0.91	1.13	36	1587	83

**Table 3 nanomaterials-14-00922-t003:** The fraction of fluorine dissolved as a result of a chemical attack with hydrogen peroxide on Nafion-TEG without Pt, Nafion-TEG-Pt/C composites, and pristine Nafion.

Sample	ω_Nafion_ (Loading)	*m*_F_/*m*_Nafion_
Nafion-TEG(1:4)-Pt/C	0.14	0.71
Nafion-TEG(1:2)-Pt/C	0.24	0.21
Nafion-TEG(1:1)-Pt/C	0.35	0.26
Nafion-TEG(1:4)	0.20	0.39
Nafion-TEG(1:1)	0.50	0.28
Nafion-TEG(4:1)	0.80	0.23
Nafion	1	0.12

**Table 4 nanomaterials-14-00922-t004:** Change in elemental composition, %wt., of some samples as a result of the chemical attack of hydrogen peroxide and the electrochemical effect of Nafion-TEG(1:1) and Nafion-TEG(1:1)-Pt/C.

Sample	ω_Nafion_	Treatment	C	O	F	S	Pt
Nafion-TEG(1:4)-Pt/C	0.14	pristine	83.48	3.51	4.82	0.36	7.83
		H_2_O_2_	76.82	3.15	8.24	0.59	11.19
Nafion-TEG(1:4)	0.20	pristine	88.6	4.30	7.62	0.31	-
		H_2_O_2_	77.20	4.90	17.30	0.61	-
		electrochemical	72.70	6.00	20.20	0.80	-
Nafion-TEG(4:1)	0.80	pristine	50.30	5.70	4200	2.00	-
		H_2_O_2_	44.11	8.39	45.20	2.30	-
		electrochemical	49.20	7.20	41.7	1.89	-
Nafion	1.00	pristine	33.02	7.07	57.70	2.22	-
		electrochemical	38.40	9.10	50.50	2.00	-

**Table 5 nanomaterials-14-00922-t005:** Raman spectra characteristics for Nafion-TEG composites before and after H_2_O_2_ treatment.

Sample	Treatment	*I*_D_/*I*_G_	*A*_D_/*A*_G_	FWHM(*G*), cm^−1^	*G*-Band Position, cm^−1^	FWHM(*2D*), cm^−1^
Nafion-TEG(1:4)	pristine	0.41	0.68	24	1579	84
	after H_2_O_2_	0.70	0.70	33	1572	88
Nafion-TEG(1:1)	pristine	0.40	0.53	33	1568	85
	after H_2_O_2_	0.64	0.75	28	1577	82
Nafion-TEG(4:1)	Pristine	0.47	0.57	29	1568	80
	after H_2_O_2_	0.55	0.77	30	1572	80

## Data Availability

Data will be made available on request.

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
