# Peer review of "Stability of Graphene/Nafion Composite in PEM FC Electrodes"

_nanomaterials, 2024, doi:10.3390/nano14110922_

Round 1

Reviewer 1 Report

Comments and Suggestions for Authors

This manuscript deals with electrochemical, chemical, and thermal resistances of the graphene/perfluorinated sulfonic acid ionomer (Nafion) composite for fuel cell electrodes. Several results reported here is important for the researchers and engineers in this field. However, the present manuscript includes several unclear points: some discussions are speculative and insufficient. In particular, the effect of additive amount of graphene on in various stabilities should be discussed in detail based on the microstructural information of the composites. My concerns are below.

 Major

1.      Title: This manuscript focus on the stability of ionomer/carbon composite in the electrodes (or catalyst layers). Therefore, I think “Stability of Graphene/Nafion composite in polymer electrolyte membrane fuel cell electrodes” is more appropriate for this work.

2.      As the reference sample for the electrode, I think that Nafion with Pt/C or with C are more suitable than pristine Nafion. What do the authors think this point?

 3.      As the authors pointed out that mechanical destruction is one of the most significant direction of degradation of ionomer (Lines 88-91 in Introduction). Why did not authors carry out mechanical stability test of the composites for FC electrodes in this comprehensive study? The authors should make this point clear.

 4.      Figure 3: For the Nafion-TEG without Pt/C samples, the m/m0 value finally reached zero. On the other hand, for the Nafioin-TEG without Pt/C samples, a small amount of sample residue remained above 700 degC (From the Table 1, the Pt content is same for all the three-component samples, therefore, this would be not from Pt.) The authors explained the presence of Pt enhance degradation of the composites (Lines 299-301), but this residue contradicts authors’ explanation. The authors should add detailed explanation concerning this point.

 5.      If possible, the electrical conductivity of the prepared Nafioin-TEG composites with and without Pt should be shown as a fundamental property.

 6.      The authors referred the dispersity of Nafion in a sample in the discussion on chemical stability. However, the detailed explanation for the term “dsipersity” and experimental data concerning the dispersity are lacking. The authors should make these points clear.

 7.      Lines 500-503: “The low value of H2 crossover current density (0.33 mA/cm2 – Nafion-TEG(1:1)-Pt/C) 500 may be associated with the formation of a barrier layer on the membrane surface during the fabrication process of the MEA, which prevents the crossover of molecular hydrogen through the membrane.”

Effect of barrier layers on H2 crossover does not depend on the G-content in the composites. The authors should add detailed discussion concerning this point.

 Minor

8.      Line 83: The authors should add the definition of C in “Nafion-C interfaces”.

9.      2. Materials: The manufacturer of Nafion dispersion DE1021 is Chemours Company, not Du Pont at present. Please confirm it.

10.  Line 148: Is it possible to describe the volume of the initial dispersions instead of 2 drops?   Several mL?

11.  Figure 5: For comparison, the range of vertical axis should be unified for all the CV curves.

Comments on the Quality of English Language

I think it good to have a Native speaker check your manuscript before resubmitting it.

Reviewer 2 Report

Comments and Suggestions for Authors

In this paper, the authors conducted a comprehensive study on the stability (electrochemical, chemical, and thermal) of fuel cell electrode composites containing few layer graphene modified additives. I believe that publication of the manuscript may be considered only after the following issues have been resolved.

1.    In order to better understand the physical mechanism of the material system, it is recommended that the author provide a schematic diagram of the fuel cell mechanism.

2.    This article provides a detailed horizontal comparison of relevant performance, that is, a comparison between oneself and oneself. Suggest the author to provide a table for a detailed comparison with the work of others.

3.    The article is missing the final section, the conclusions section.

4.    In order to increase the readability of the article, in the introduction section, the author needs to mention some relevant latest references on the application of graphene-based material, such as, Diamond and Related Materials 142, 2024, 110793; Opto-Electron Sci 2, 230012 (2023); Opto-Electron Adv 5, 200098 (2022).

5.    The English expression of the whole article needs to be further improved.

Comments on the Quality of English Language

Minor editing of English language required

Round 2

Reviewer 1 Report

Comments and Suggestions for Authors

The revised manuscript should be accepted.

Reviewer 2 Report

Comments and Suggestions for Authors

Accept in present form